# Role of Mg(NO_3_)_2_ as Defective Agent in Ameliorating the Electrical Conductivity, Structural and Electrochemical Properties of Agarose–Based Polymer Electrolytes

**DOI:** 10.3390/polym13193357

**Published:** 2021-09-30

**Authors:** N. I. Ali, S. Z. Z. Abidin, S. R. Majid, N. K. Jaafar

**Affiliations:** 1Faculty of Applied Sciences, Universiti Teknologi MARA, Shah Alam 40450, Selangor, Malaysia; niaizzati2562@gmail.com (N.I.A.); norka603@uitm.edu.my (N.K.J.); 2Ionic Materials and Devices (iMADE) Research Laboratory, Institute of Science, Universiti Teknologi MARA, Shah Alam 40450, Selangor, Malaysia; 3Centre for Ionics University of Malaya, Department of Physics, Faculty of Science, University of Malaya, Kuala Lumpur 50603, Malaysia

**Keywords:** agarose, magnesium nitrate, polymer electrolyte, conductivity, dielectric, FTIR, XRD, electrochemical

## Abstract

Polymer electrolytes based on agarose dissolved in DMSO solvent complexed with different weight percentages of Mg(NO_3_)_2_ ranging from 0 to 35 wt% were prepared using a solution casting method. Electrochemical impedance spectroscopy (EIS) was applied to study the electrical properties of this polymer electrolyte, such as ionic conductivity at room and different temperatures, dielectric and modulus properties. The highest conducting film has been obtained at 1.48 × 10^−5^ S·cm^−1^ by doping 30 wt% of Mg(NO_3_)_2_ into the polymer matrix at room temperature. This high ionic conductivity value is achieved due to the increase in the amorphous nature of the polymer electrolyte, as proven by X-ray diffractometry (XRD), where broadening of the amorphous peak can be observed. The intermolecular interactions between agarose and Mg(NO_3_)_2_ are studied by Fourier transform infrared (FTIR) spectroscopy by observing the presence of –OH, –CH, N–H, CH_3_, C–O–C, C–OH, C–C and 3,6-anhydrogalactose bridges in the FTIR spectra. The electrochemical properties for the highest conducting agarose–Mg(NO_3_)_2_ polymer electrolyte are stable up to 3.57 V, which is determined by using linear sweep voltammetry (LSV) and supported by cyclic voltammetry (CV) that proves the presence of Mg^2+^ conduction.

## 1. Introduction

A significant increase in interest can be seen in the research activities for formulating suitable materials for next-generation advanced rechargeable batteries by virtue of high energy density in theory [1]. Multivalent-ion batteries such as Mg^2+^, Ca^2+^ and Al^3+^ are reported to deliver higher volumetric energy densities compared with monovalent-ion batteries, including Li-ion and Na-ion batteries. Among these promising multivalent ion batteries, Mg-ion batteries represent a significant advance in battery technology due to their highest theoretical volumetric energy density (3866 mAh cm^−3^), which is higher than that achievable by a Li anode (2066 mAh cm^−3^) [2,3,4]. However, several challenges have limited the progress of Mg-ion battery technology maturation [3,4,5,6], including Mg-ion insulating passivation layer on the anode surface [7], the solid electrolyte interface (SEI) [8], safety and stability, and their narrow electrochemical operation window (<2 V vs. Mg/Mg^2+^) [4,9].

Research on amorphous polymer-salt electrolytes is widely reported due to their wide variety of applications in devices such as fuel cells, supercapacitors and electrochromic windows [8,10]. This type of electrolyte offers high safety, lower cost, straightforward processability and good handleability during cell fabrication which are not discovered in liquid electrolytes [11,12]. The crucial part of producing a high-performance battery is the selection of suitable electrolytes to match with the magnesium electrode. Liquid electrolytes are commonly used in the production of batteries as they can exhibit excellent ionic conductivity [13], but they were found to have some problems due to their properties, such as easily leakage and causing corrosion during packaging [14,15,16]. Aside from that, most liquid electrolytes are highly toxic and combustible [13,17], so they can contaminate the environment and can also harm consumers. These facts have led to the innovation and development of polymer electrolytes for the production of rechargeable batteries. Polymer electrolytes can reliably overcome the risk of leakage and offer good mechanical properties, good stability at the electrode–electrolyte interface and they can withstand high temperatures [18].

Earlier studies on biopolymers as amorphous polymer salt complexes have shown that these naturally abundant polymers exhibited good film producibility [19]. Collective claims on biopolymers as a host for ion conduction include cellulose, starch, and carrageenan, which have room temperature conductivities in the range of 10^−5^–10^−4^ S·cm^−1^ [13,14,18]. Agarose is a seaweed-derived biopolymer employed as a host polymer for Mg^2+^ conduction. The Mg^2+^ conduction is supported by the hydroxyl groups in the repeating units of agarose, which contain an electronegative atom (oxygen). The presence of oxygen atoms also results in the formation of a cross-linking network to other constituents that can promote the transportation of ions in the polymer matrices [20,21]. The success of agarose as a stable polymer electrolyte gel film with an ionic conductivity almost similar to those of ethylene carbonate (EC), propylene carbonate (PC) and lithium trifluoromethanesulfonate (LiCF_3_SO_3_) liquid electrolyte that reached 2.80 × 10^−3^ S·cm^−1^ that has been recorded by Perera and Dissanayake [20,22]. Singh and co-workers reported an ionic conductivity of 9.02 × 10^−3^ S·cm^−1^ achieved when 40 wt% potassium iodide (KI) was infused in agarose matrices [22]. A study on agarose–lithium iodide (LiI) has reported an optimum ionic conductivity value of 3.98 × 10^−3^ S·cm^−1^ when doped with 20 wt% LiI [23]. A particular focus of documented conductivity studies has centred at the ion charge carriers generated by the doped salts that produce more ion carriers for conduction in the electrolytes. The concentration and lattice energy of dopant salts, as well as the surrounding temperature, are generally reported to have substantial effects on the conductivity and dielectric relaxation of the electrolytes [24,25]. These two aspects allow the researcher to unravel the valuable behaviour of cation-polymer interactions, including salt dissolution, infusion and ion transports which are associated with the structural nature of the electrolytes.

The present work was carried out to investigate the effect of Mg^2+^ on the conductivity, dielectric and complexation nature in agarose–Mg(NO_3_)_2_ as a freestanding film prepared by means of a solution casting method. Mg(NO_3_)_2_, as dopant salt with a high lattice energy of 2429.51 kJ·mol^−1^ [26], was chosen to minimise the occurrence of ion association between cations and anions in the complexes. The results revealed in the current work indicate that divalent salts with moderate conductivity can stimulate the development of polymer electrolytes for multivalent batteries [4].

## 2. Experimental

### 2.1. Materials

Agarose (molecular weight, M_w_: 630.5 g·mol^−1^) obtained from Next Gene (Puchong, Malaysia) was used as host polymer, and magnesium nitrate salt (Mg(NO_3_)_2_; M_w_: 256.41 g·mol^−1^, ACS reagent 99%) purchased from Sigma-Aldrich (Saint Louis, MI, USA) was used as a dopant. Dimethyl sulfoxide (DMSO) with ≥99.7% purity and 78.129 g·mol^−1^ molecular weight was purchased from Fisher Scientific (Hampton, NH, USA) to function as a solvent in the preparation of polymer electrolyte films.

### 2.2. Preparation of Polymer Electrolyte Films

A solution casting method was used to prepare agarose–Mg(NO_3_)_2_ polymer electrolyte films. Thus agarose powder (0.5 g) was dissolved in dimethyl sulfoxide (DMSO, 20 mL) before different amounts of Mg(NO_3_)_2_ was added to the solution. The Mg(NO_3_)_2_ ranged from 0 to 35 weight percent (wt%) with 5 wt% intervals. The solution of agarose–DMSO–Mg(NO_3_)_2_ was stirred magnetically until the solution became clear and homogeneous. Then, the samples were cast into a clean Petri dish and left to dry until a colourless thin film of polymer electrolyte was formed. The electrolyte films were tagged as AMg0, AMg5, AMg10, AMg15, AMg20, AMg25, AMg30 and AMg35 for 0, 5, 10, 15, 20, 25, 30 and 35 wt% Mg(NO_3_)_2_ infused into the agarose matrices, respectively.

### 2.3. Characterisation Techniques

#### 2.3.1. Electrochemical Impedance Spectroscopy (EIS)

Electrochemical impedance spectroscopy (EIS) was performed using a HIOKI 3532-50 LCR Hi-Tester at various temperatures ranging from 300 to 373 K. The samples were sandwiched between two symmetrical stainless-steel electrodes with a contact area of 0.7854 cm^2^. During the measurements, the frequency was fixed in the range of 100 Hz to 1 MHz. From this measurement, the value of the bulk impedance, R_b_ is determined from the plot of negative imaginary impedance, -Zi versus the real part, Zr of impedance and used to calculate the ionic conductivity of agarose-based polymer electrolyte by using the following formula:(1)$$\sigma =\frac{t}{R_{b}A}$$
where *σ* (S·cm^−1^) is the ionic conductivity of polymer electrolyte, A (cm^2^) is the electrode–electrolyte contact area, t (cm) is the thickness of the thin film, and R_b_ is the bulk impedance. Other parameters related to the ion conduction mechanism, i.e., dielectric and modulus, were also plotted and discussed in this work.

#### 2.3.2. Fourier Transform Infrared (FTIR) Spectroscopy

The molecular interactions in different compositions of Mg(NO_3_)_2_ in agarose–based polymer electrolytes were analysed by FTIR spectroscopy using Perkin Elmer (Waltham, MA, USA) model spectrum 400 instrument. The spectra were recorded in the absorbance mode in the wave region ranging from 650 cm^−1^ to 4000 cm^−1^ with a spectral resolution of 2 cm^−1^. This measurement was conducted at room temperature by means of an attenuated total reflection (ATR) device.

#### 2.3.3. X-ray Diffraction (XRD)

X-ray diffraction (XRD) patterns were measured using an PANalytical X’pert PRO diffractometer with CuK_α_ radiation (λ = 1.5418 Å) to determine the nature of polymer electrolyte formed either crystalline or amorphous. In XRD measurement, the Bragg angle, 2θ was varied from 5° to 90° at room temperature (27 °C).

#### 2.3.4. Electrochemical Characterisation

The electrochemical stability of polymer electrolytes was studied by linear sweep voltammetry (LSV) using an Automatic Battery Cycler (WBCS 3000, WonA Tech, Seoul, South Korea). The highest conducting agarose–Mg(NO_3_)_2_ polymer electrolyte that would act as a separator was placed in between stainless steel (SS) and magnesium metal (Mg). The stainless steel served as a working electrode and the magnesium metal as a reference/counter electrode. LSV was performed at room temperature at a fixed potential range from 0 to 4 V at a scan rate of 5 mV·s^−1^. The cyclic voltammetry (CV) was performed in two-electrode configurations at a scan rate of 5 mV·s^−1^ in the voltage range of −1.5 V to 3.5 V. By carrying out this characterisation, the presence of the redox reactions in agarose-based polymer electrolyte can be discussed.

## 3. Results and Discussions

### 3.1. Impedance Analysis

The impedance spectra of selected agarose–Mg(NO_3_)_2_ electrolyte films presented in Figure 1a–d consist of the imaginary part versus the real part of the impedance. The important feature that can be highlighted from these spectra is the mechanism of charge transport in the film samples in the high frequencies region, which can be used effectively in the determination of DC conductivity, σ_DC_ value. Commonly, a semicircle at the higher frequency in the impedance spectra is recorded, which refers to the ion transport through the bulk electrolyte with a particular ionic conductivity [24,27,28]. In the present work, there is no distinguishable spike in the low frequencies region as seen from the impedance spectrum of pure agarose film (Figure 1a), which proved the absence of charge carriers that can take part in ion diffusion when the electrode polarisation (EP) has taken place. EP is related to the electric double-layer capacitance that arises from charges accumulation at the oppositely charged electrodes and leads to the distribution of dielectric relaxation times in polymer electrolyte films [24]. Although this phenomenon may hinder the impedance measurements, the R_b_ value is successfully determined using a manual graphical approach for pure agarose films to calculate the conductivity. The conductivity of pure agarose film is 1.75 × 10^−8^ S·cm^−1^. However, as seen clearly from Figure 1b–d, a protruding 45° angular straight line appears in the low frequencies region in the salt-added agarose films, indicating that the diffused ion charge carriers actively participate in the electrode polarisation (EP) process [25]. In this case, the intercept of a spike at the *x*-axis with the end-tail of the semicircle is used to determine the R_b_ [21] and calculate the ionic conductivity of agarose–Mg(NO_3_)_2_ polymer electrolyte at various weight percentages (wt%) of Mg(NO_3_)_2_ as displayed in Figure 2a. The ionic conductivity increased to 1.13 × 10^−6^ S·cm^−1^ upon addition of 25 wt% Mg(NO_3_)_2_ and further increased to 1.48 × 10^−5^ S·cm^−1^ when 30 wt% Mg(NO_3_)_2_ is dissolved into the agarose matrix. The enhancement of ionic conductivity is attributed to the higher number of ion dissociation events obtained as a consequence of the addition of salt Mg(NO_3_)_2_ into the agarose-based polymer electrolyte system that results in the production of free ions (Mg^2+^ cation and NO_3_^−^ anion) that can be incorporated in the polymer host matrix [29]. Therefore, a high salt concentration promotes the degree of ion dissociation, thus increasing the ionic conductivity of the agarose–Mg(NO_3_)_2_ polymer electrolyte system. On the other hand, the increase in amorphous nature of the system also affects the enhancement of ionic conductivity as it leads to flexibility of the agarose polymer chain and promotes the segmental motion in this system as represented in Figure 2b [21,29]. However, the ionic conductivity starts to decrease at 35 wt% Mg(NO_3_)_2_ with the value of 2.47 × 10^−6^ S·cm^−1^ due to the aggregation of ions, which leads to the formation of ion cluster carriers or overcrowded ions that cause a decline in both the number of ions and their mobility [30,31]. Furthermore, the overabundance of ions in the host polymer might cause ionic contact with the polymer chains, restricting the polymer chains’ segmental relaxation [32]. Formerly documented conductivities of *i*-carrageenan-Mg(NO_3_)_2_ [33] and polyethylene oxide (PEO)-Mg(NO_3_)_2_ [34] polymer electrolytes at 303 K are 6.10 × 10^−4^ S·cm^−1^ and 1.34 × 10^−5^ S·cm^−1^, respectively.

Generally, the conductivity is related to the ion charge carriers concentration (n_k_) and ions mobility (µ_k_) which can be mathematically described from the equation:(2)$$\sigma =\sum n_{k}q_{k}\mu_{k}$$
where, q = the electron charge.

The proportionality of conductivity (*σ*) with charge carrier concentration and/or ions mobility is directly explained by the influence of salt concentration on the conductivity. According to the past reported conduction mechanism of polymer-salt systems, the independent charge transport species from the disjointed cation-anion move under an applied electric field. The cation leaps from one electronegative atom to another, creating a vacancy that another readily available cation present can fill at the neighbouring atom. This repetitive leaping of charge transport species is known as the Grotthus mechanism and is related to cation-anions disjointed via a dissociation process [35,36]. The effectiveness of salt dissociation can be improved by using a salt with a low lattice energy and relying on the cation size [23,32]. The lattice energy of Mg(NO_3_)_2_ is 2429.51 kJ·mol^−1^ [26] and is presumed to exhibit weak salt dissociation, resulting in low room temperature conductivity. Using the same relationship, the conductivity can be said to be affected by the temperature. Figure 2c illustrates the graph log *σ* vs. 1000/T for agarose–Mg(NO_3_)_2_ polymer electrolytes with selected salt concentrations. Here, the conductivity of polymer electrolyte films is linearly dependent of the temperature, and this relation can be related to the Arrhenius rule as given in the following equation:(3)$$\sigma =\sigma_{0}\exp\frac{-E_{a}}{k_{B}T}$$
where *σ*_0_ is the pre-exponential factor, *E_a_* is the activation energy, and *k_B_* is the Boltzmann constant. The supplied thermal energy weakens the recombination of ions during the solvent desiccation step in the film formation. Therefore, more independent ions are available for conduction as the temperature increases, contributing to higher conductivity values. The activation energy of the selected AMg25 and AMg30 samples are determined as 0.142 eV and 0.044 eV, respectively. The highest conducting of agarose–Mg(NO_3_)_2_ polymer electrolyte, AMg30, the activation energy has recorded a low value implying that low energy is required to free the ions hopping to another coordinating site [33,37]. The typical polymer host used in electrolyte fabrication is semicrystalline, characterized by displaying crystalline and amorphous domains in the structure. The latter domains are claimed to be the most favourable for ion conduct conduction media. Therefore, heat built up with the increasing temperature generates larger disordered domains keen for intra- and inter-chain ion hopping through internal bond rotations that cause the polymer segmental motion that grants the conductivity enhancement [8,10].

### 3.2. Dielectric Study

Dielectric studies were performed to get further information on the conductivity behaviour of the studied divalent ion-based electrolyte. The complex permittivity (*ε**) consist of the real and imaginary part of dielectric, and their relationship is as in the equation below:(4)$$\varepsilon^{*}=\varepsilon_{r}-i\varepsilon_{i}$$

The real component, $\varepsilon_{r}$ explains the store charge in the material and is expressed by:(5)$$\varepsilon_{r}=\frac{Z_{i}}{\omega C_{0}\left(Z_{r}^{2}+Z_{i}^{2}\right)}$$

The imaginary component (dielectric loss), $\varepsilon_{i}$ represents the components of energy loss of mobile charge when the polarity of the electric field is reversed [38] as shown in the following equation:(6)$$\varepsilon_{i}=\frac{Z_{r}}{\omega C_{0}\left(Z_{r}^{2}+Z_{i}^{2}\right)}$$
where, $C_{0}=\varepsilon_{0}A/t$ and $\varepsilon_{0}$ = the permittivity of free space (8.854 × 10^−12^ F·m^−1^). The angular frequency, ω=2πf, $Z_{r}$ = the real impedance and $Z_{i}$ = the imaginary impedance. Figure 3a shows the plot of the dielectric constant of 30 wt% Mg(NO_3_)_2_ at various temperatures. It indicates that $\varepsilon_{r}$ is slowly decreasing as the frequency increases before the value becomes almost stagnant at high frequencies due to the electrode polarisation. At low frequency, $\varepsilon_{r}$ rises sharply due to the mobile charges that migrate along the field before being framed by blocking electrodes and causing the accumulation of ions at the electrode–electrolyte interface to form an electrical double layer [38]. In addition, the built-up charges have sufficient time to sustain the polarisation and contribute to the higher value of the dielectric constant in the low-frequency region. The higher the dielectric constant, the higher the fractional number of mobile charges. The dielectric constant also increases with the temperature as can be corroborated by increasing the number of free ions. For the high-frequency region, the value of the dielectric constant decreases and reaches a plateau due to the fast rate of the periodic reversal process of the electric field direction causing a reduction of the polarisation effect as no charge is built up at the electrode–electrolyte interface. Even at higher temperatures (343 and 363 K), the polarisation effect is discontinued due to the random motion of mobile charges that causes imperfect polarisation to occur [31]. Figure 3b evidences the increased $\varepsilon_{i}$ in the low frequency region and further increases with the temperature. The observation infers the production of a larger number of mobile charges that promote the interaction between random mobile ions hence causing a longer relaxation time. In the high-frequency region, rapid reversal of the electric field leads to deficiencies in charge diffusion and low $\varepsilon_{i}$ values [31]. The further dielectric analysis is further parameterised by loss tangent (tan *δ*) and the relaxation time (τ), determined using the following relations:(7)$$\tan\delta =\frac{\varepsilon_{i}}{\varepsilon_{r}}$$
(8)$$2\pi f_{max}\tau =1$$

Figure 3c depicts tan *δ* as a function of the logarithm of frequency (f) of agarose–30 wt% Mg(NO_3_)_2_ polymer electrolyte at different temperatures. The temperatures stimulate the frequencies position and the intensities of the peak in the plots. Such stimulation authenticated the dielectric relaxation process that is thermally activated. The relaxation time (τ) explains how the ionic charges carrier in materials align with the direction of the applied field, and at a higher temperature, the longest τ is achieved. In high temperature surroundings, easily moved ions are produced that take part in conduction and attain relaxation at the higher frequency [39].

Further insights in the relaxation process can be acquired from the electric modulus where the real part of the modulus ($M_{r}$) and the imaginary part of the modulus ($M_{i}$) are expressed as below:(9)$$M_{r}=\frac{\varepsilon_{r}}{\left[{\left(\varepsilon_{r}\right)}^{2}+{\left(\varepsilon_{i}\right)}^{2}\right]}$$
(10)$$M_{i}=\frac{\varepsilon_{i}}{\left[{\left(\varepsilon_{r}\right)}^{2}+{\left(\varepsilon_{i}\right)}^{2}\right]}$$
where $\varepsilon_{i}$ is the dielectric loss and $\varepsilon_{r}$ is the dielectric constant of a sample.

Figure 3d,e shows the plot of the electric modulus against frequency for ($M_{r}$) and ($M_{i}$) at elevated temperatures. One can notice that a low electric modulus was recorded for the low-frequency regime, which is attributed to the concealed electrode polarisation effect [31,40]. Differently, for the electric modulus in the high-frequency regime both the real and imaginary electric modulus increase without well-defined peak formation, and the increased values are due to the bulk effect of the electrolyte samples [36]. In other aspects, the increase of temperature results in the inclination of $M_{r}$ values implying a higher movement rate of the charge carriers.

### 3.3. Fourier Transform Infrared (FTIR) Spectroscopy

Referring to the molecular structure of agarose, multiple oxygen atoms bear lone pair electrons for the conduction of cations [35]. Therefore, FTIR spectroscopy was applied to identify the interactions between Mg-ions and the oxygen atoms that can indirectly substantiate the conduction behavior in the sample. The intermolecular interactions present in the agarose–Mg(NO_3_)_2_ electrolyte system are displayed in Figure 4a,b. The significant FTIR peaks of agarose in Figure 4a centred at 3357 cm^−1^, 2900 cm^−1,^ and 1641 cm^−1^ can be ascribed to the O–H stretching vibration [41], –CH stretching [42] and N–H bending vibration modes [39], respectively.

Other peaks belonging to agarose are the CH_3_ bond bending of an alkene group, C–O–C, C–OH and 3,6–anhydrogalactose bridge vibration modes that are registered at 1368 cm^−1^ [42], 1152 cm^−1^ [43], 1064 cm^−1^ [23] and 929 cm^−1^ [43], respectively. The FTIR spectra in Figure 4b show the interactions that take place in agarose with different wt% of Mg(NO_3_)_2_. From the figure, the addition of salt shifts the OH stretching vibration peak of agarose to the 3397–3387 cm^−1^, region (i). The peak observed at 2930 cm^−1^ attributable to –CH stretching vibrations has shifted to 2922 cm^−1^ due to the increase of Mg(NO_3_)_2_ salt in the polymer matrix, resulting in higher energy vibration. At the same time, the N–H and CH_3_ bending in this work are detected at 1646–1652 cm^−1^ and 1364–1332 cm^−1^, respectively. The wavenumbers of C–O–C stretching vibration modes have been relocated from 1152 cm^−1^ to 1154 cm^−1^, proving the presence of carbohydrates units in the agarose-based polymer electrolyte [23,44,45]. A strong vibration of C–OH in agarose–Mg(NO_3_)_2_ polymer electrolyte peak is recorded at 1070 cm^−1^ to 1072 cm^−1^ as observed in the region (vii). The vibration peak of the 3,6–anhydrogalactose bridge from the agarose polymer, has remained at a wavenumber of 930 cm^−1^ [41,43]. For pure Mg(NO_3_)_2,_ as shown in the figure, we can verify the presence of the plane deformation modes of NO_3_^−^ at the peak of 819 cm^−1^, as mentioned by a work of Manjuladevi and his team for a peak between 818–820 cm^−1^ [33,46,47,48,49]. However, this peak is absent in the agarose–Mg(NO_3_)_2_ polymer electrolyte spectra.

Figure 5 depicts the FTIR deconvolution for all agarose–Mg(NO_3_)_2_ polymer electrolyte samples (AMg0, AMg5, AMg10, AMg15, AMg20, AMg25, AMg30 and AMg35) in the 1320 cm^−1^ to 1450 cm^−1^ spectroscopic region. A band at a wavenumber of 1435–1438 cm^−1^ represents the isolation of C-C stretching vibrations for every composition of the agarose-Mg(NO_3_)_2_ polymer electrolyte [50].

Further, the free ions and contact ions percentages were calculated from the following formulas:(11)$$\mathrm{Free}\;\mathrm{ion}\;\left(\%\right)=\frac{A_{f}}{A_{f}+A_{c}}\times 100\%$$
(12)$$\mathrm{Contact}\;\mathrm{ion}\;\left(\%\right)=\frac{A_{c}}{A_{f}+A_{c}}\times 100\%$$
where $A_{f}$ is an area under the peak of the free ion region and $A_{c}$ is the area under the peak of the contact ion region. In this work, the free ions peak is observed at 1404–1410 cm^−1^ which is important to clarify the conductivity trend in the sample [36]. The contact ion or ion pairs (Mg^2+^------NO_3_^−^) is effected in the polymer electrolyte due to the electrostatic forces that form between oppositely charged ions in the electrolyte. Another type of coexistent species is the ion aggregates (Mg^2+^------NO_3_^−^------Mg^2+^) that originate from the interactions among the Mg(NO_3_)_2_ units in the agarose matrix and are assumed to be larger than ion pairs. The vibrational frequencies encountered at 1419–1423 cm^−1^ and 1340–1366 cm^−1^ are assignable to ion pairs and ion aggregates, respectively [36]. From these decomposed peaks, the area peak of free ions in AMg0 is lower than in AMg5 due to the absence of dopants in the polymer matrix. After adding about 5 wt% of Mg(NO_3_)_2_ to the agarose based electrolyte, the area of the free ions peak starts to increase and this causes a decrease of the area under the peak of the contact ions region as the salt begins to dissociate in this composition. The samples of polymer electrolyte labelled as AMg10, AMg15, AMg20, and AMg25 record an area under the peak of free ions that is almost similar to that of the the sample of AMg5 as these concentrations exhibit a comparable ionic conductivity value, which is about 10^−7^ to 10^−6^ S·cm^−1^. However, the band attributed to free ions has shifted from 1408 cm^−1^ in AMg0 to 1404 cm^−1^ in AMg25, while for AMg30, the area under the free ions peak is larger than the peak area of the contact ions, which reveals that the lowest percentage of contact ions is achieved in the highest conducting agarose–30 wt% Mg(NO_3_)_2_ polymer while the percentage of free ions is the highest, reflecting the occurrence of a high degree of ion dissociation in the sample as illustrated in Figure 6. However, the decrease in the percentage of free ions and rise in the percentage of ion pairs is observed upon adding 35 wt% of Mg(NO_3_)_2_, induced by ion recombination to form ion pairs. A plausible mechanism for the complexation between agarose molecules and Mg(NO_3_)_2_ is illustrated in Figure 7, where the magnesium (Mg^2+^) and nitrate (NO_3_^−^) ions are coupled to the ether oxygens of agarose. The Mg^2+^ are more tightly bound within the matrix, while the NO_3_^−^ are loosely attached. In the presence of a small DC electric field, the weakly bound NO_3_^−^ can rapidly dissociate and are able to hop from one coordinating site to another.

### 3.4. XRD Analysis

The semicrystalline properties nature of the polymer electrolyte has been studied by using X-ray diffractometry (XRD). A substantial defect formation effect in the semicrystalline matrix increases the amorphous domain that generally improve the ion conduction for conductivity enhancement. The role of the defect agent Mg(NO_3_)_2_ in the agarose molecules has been highlighted in the FTIR as evidenced by the shifting of OH, –CH, N–H, –CH_3_, C–O–C, C–OH, C–C vibrational modes. XRD diffraction peaks of pure salt, Mg(NO_3_)_2_ are observed at 2θ = 20.12°, 26.99°, 29.56°, 38° and 43.53° as illustrated in Figure 8a. These angles are in accordance with the study done by Priya and her team, which recorded the diffraction peak of the same salt at 2θ angle of 20.542°, 26.667°, 29.257°, 38.100° and 43.692° [33].

However, these peaks are absent in the XRD pattern of agarose–Mg(NO_3_)_2_ polymer electrolytes shown in Figure 8b, which indicates the salt has wholly dissociated when incorporated in the agarose matrix [46]. The presence of amorphous peaks confirms the semicrystalline nature of agarose molecules at 2θ = 20.2° at 0 wt% concentration of Mg(NO_3_)_2_ in a polymer matrix (AMg0) [22]. The addition of Mg(NO_3_)_2_ to the polymer matrix results in the peak becomes broader as can be observed from the XRD patterns of AMg5 to AMg30, thus, proving the increased amorphous nature of the polymer electrolyte system, that causes the high dissociation of the salt into free ions and increases the migration of ions through the polymer chains. In AMg35, the peak becomes narrow, indicating a decrease in the amorphous nature of the electrolyte due to the reassociation of ions in the polymer matrix.

These observations are confirmed through the XRD deconvolution technique, which is analysed using a Gaussian function to identify the exact position of peaks that appear in selected samples of agarose–Mg(NO_3_)_2_ polymer electrolyte system and represent the trend of amorphousness nature as illustrated in Figure 9. Furthermore, this technique also be able to determine the degree of crystallinity, $X_{c}$ and crystalline size, L of the electrolyte and evaluated by using Equations (13) and (14), respectively [51]:(13)$$X_{c}=\frac{A_{c}}{A_{c}+A_{a}}\times 100\%$$
where $A_{c}$ is area under the peak of crystalline region and $A_{a}$ is the area under the peak of amorphous region:(14)$$L=\frac{0.9\lambda }{FWHM\cos\theta }$$
where *L* is the crystallite size, λ is the X-ray wavelength, which is fixed at (1.5406 Å), *FWHM* is the full width at half maximum (a measure of the peak broadness) and *θ* is Bragg’s diffraction angle.

From the XRD deconvolution pattern for electrolyte shown in Figure 9, the amorphous peak broadening due to the increase of *FWHM* value leads to the reduction of crystallite size from 6.62 nm to 3.17 nm, as presented in Table 1. The peaks at 2θ = 22.7° (AMg0 and AMg5), 23.0° (AMg20), 23.0° (AMg30) and 22.8° (AMg35) refer to the peaks of the crystalline domain. As observed, the crystalline peak area decreases in each sample from AMg5 to AMg30, further supporting the increment in the electrolyte’s amorphous nature. The amorphous peak intensity has increased and shifted to 2θ = 20.2° in sample AMg30, which reveals it as having the highest amorphous nature among the agarose-based polymer electrolytes as supported by its highest ionic conductivity value. Therefore, this composition exhibits the lowest degree of crystallinity and crystallite size, which are recorded at 11.29% and 3.03 nm, respectively. In AMg35, the amorphous peak becomes narrow and decreases in intensity due to the reduction of the electrolyte’s amorphous nature, which is recorded at about 37.94%. On the other hand, the crystallite size becomes more significant up to 3.17 nm. These results are in accord with the trend of ionic conductivity obtained in the electrochemical impedance spectroscopy (EIS) experiments.

### 3.5. Voltage Stability

Figure 10a shows the LSV of agarose–30 wt% of Mg(NO_3_)_2_ in the SS//AMg30//Mg configuration. The current is swept from 0 V towards the positive range to determine the stability of the polymer electrolyte film that is important for further practical application. The voltage stability ceases when the cell does not withstand the potential. At this point, a rapid increase in current occurs due to the decomposition of the electrolyte at the inert electrode interface, known as the onset current. The onset current is marked at 3.57 V, inferring the stability of agarose–30 wt% of Mg(NO_3_)_2,_ which is a good range for electrochemical applications in magnesium batteries [38,52]. Manjuladevi et al. reported PVA:PAN/Mg(NO_3_)_2_ exhibits voltage stability of 3.4 V with the conductivity of 1.7 × 10^−3^ S·cm^−1^ [46]. In another work Aziz and co-workers reported a breakdown voltage of 2.2 V when magnesium was used as dopant salt in the electrolyte fabrication [53].

The cyclic voltammetry (CV) pattern of the highest conducting sample of agarose-based polymer electrolyte, AMg30, that had been sandwiched between a stainless steel and a magnesium electrode in the range of 0 to 3.5 V is illustrated in Figure 10b. The CV pattern has been divided into the small scale in a range of 0 to 2 V to observe the anodic and cathodic current peaks as shown in Figure 10c, whereas the anodic peaks at 0.4 V and 1.2 V while cathodic peaks appear at 0.7 V and 1.4 V. The presence of these peaks proves the existence of the Mg^2+^ conduction in the polymer electrolyte system [18].

## 4. Conclusions

Agarose–based polymer electrolytes shows their amorphous nature behaviour with the addition of 30 wt% of Mg(NO_3_)_2_ and exhibit the highest conductivity at a value of 1.48 × 10^−5^ S·cm^−^^1^ and a lowest activation energy value of about 0.044 eV due to the higher degree of ion dissociation. Beyond 30 wt% of salt in the polymer matrix, the ionic conductivity becomes reduced as the aggregation of ions starts to occur. The correlation of ionic conductivity with temperature can be explained by an enhancement of the mobility of ions at high temperatures, thus promoting the ionic conductivity of agarose–Mg(NO_3_)_2_ polymer electrolyte. FTIR studies show the presence of ν(O–H), ν(–CH), δ(N–H), δ(CH_3_), ν(C–O–C), ν(C–OH), 3,6–anhydrogalactose bridge vibrations and ν(C–C) in the agarose–based polymer electrolyte. The structural behaviour of this electrolyte proves that the peak formed in agarose polymer around 18.8° becomes broader when the amount of salt incorporated in the polymer matrix increases and results in the decrement in the constituents’ complexation. Therefore, this proves that the highest conducting film of agarose–Mg(NO_3_)_2_ polymer electrolyte is in accordance with its intermolecular and amorphous behaviour. Furthermore, the agarose–Mg(NO_3_)_2_ electrolyte system is proven to exhibit excellent electrochemical properties as it can attain a sufficient range of electrochemical stability up to 3.57 V, which makes it applicable in magnesium batteries.

## Figures and Tables

**Figure 1 polymers-13-03357-f001:**
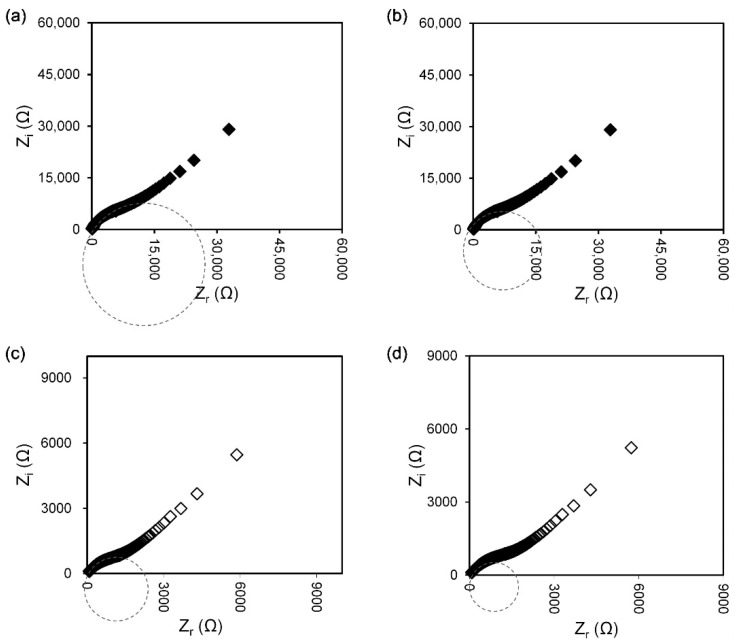
Nyquist plots for (**a**) AMg0, (**b**) AMg25, (**c**) AMg30, (**d**) AMg35 polymer electrolyte films.

**Figure 2 polymers-13-03357-f002:**
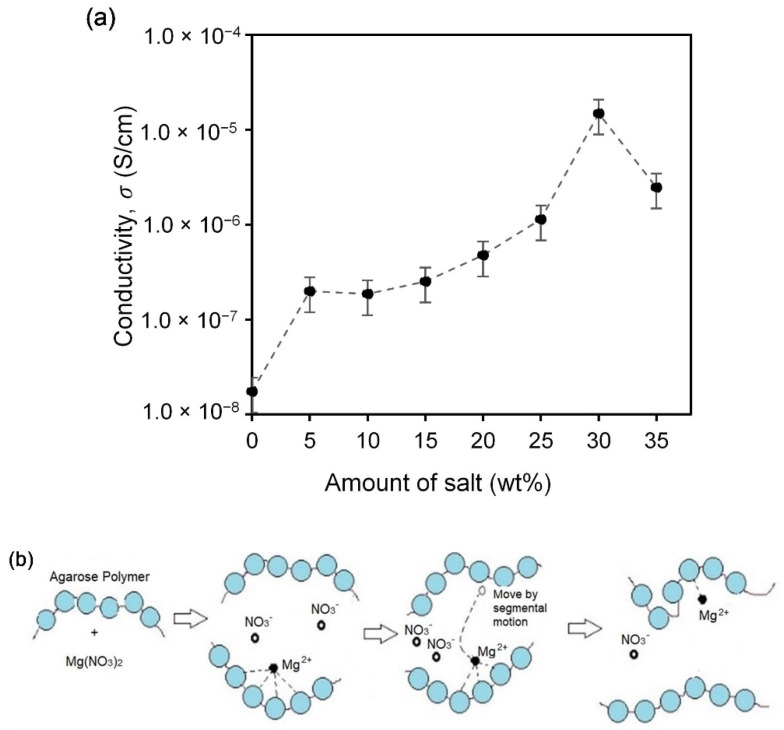
(**a**). Graph of ionic conductivity of agarose‒Mg(NO_3_)_2_ in room temperature, (**b**) schematic of segmental motion of Mg^2+^ cation into the polymer chain and (**c**) ionic conductivity in various temperatures for the selected composition of agarose‒Mg(NO_3_)_2_ polymer electrolyte.

**Figure 3 polymers-13-03357-f003:**
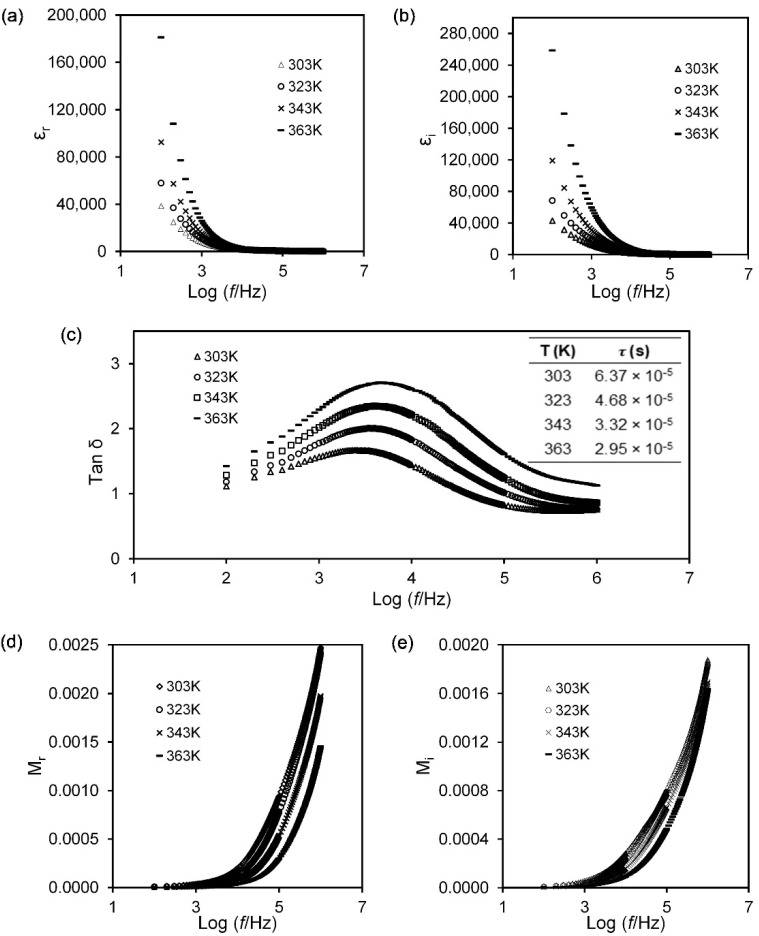
Graph of (**a**) dielectric constant, (**b**) dielectric loss, (**c**) tan *δ* with table of relaxation value, (**d**) real modulus, (**e**) imaginary modulus as a function of the logarithm of frequency for sample AMg30 at selected temperatures.

**Figure 4 polymers-13-03357-f004:**
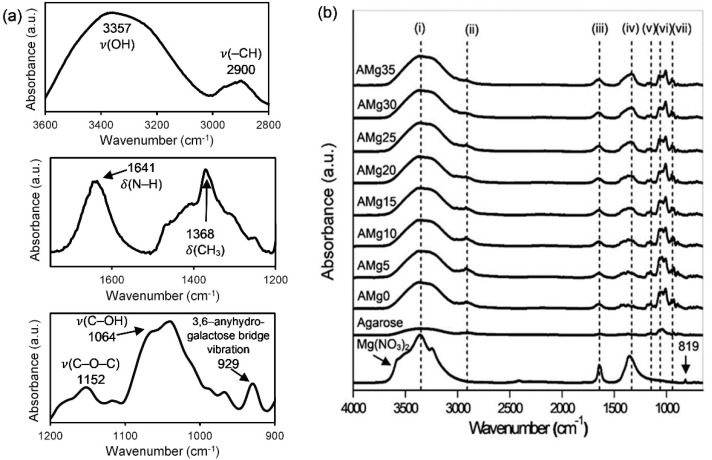
FTIR spectra of (**a**) pure agarose in region 3600–2800 cm^−1^, 1750–1200 cm^−1^ and 1200–900 cm^−1^ and (**b**) Mg(NO_3_)_2_, pure agarose and agarose–Mg(NO_3_)_2_ polymer electrolyte film.

**Figure 5 polymers-13-03357-f005:**
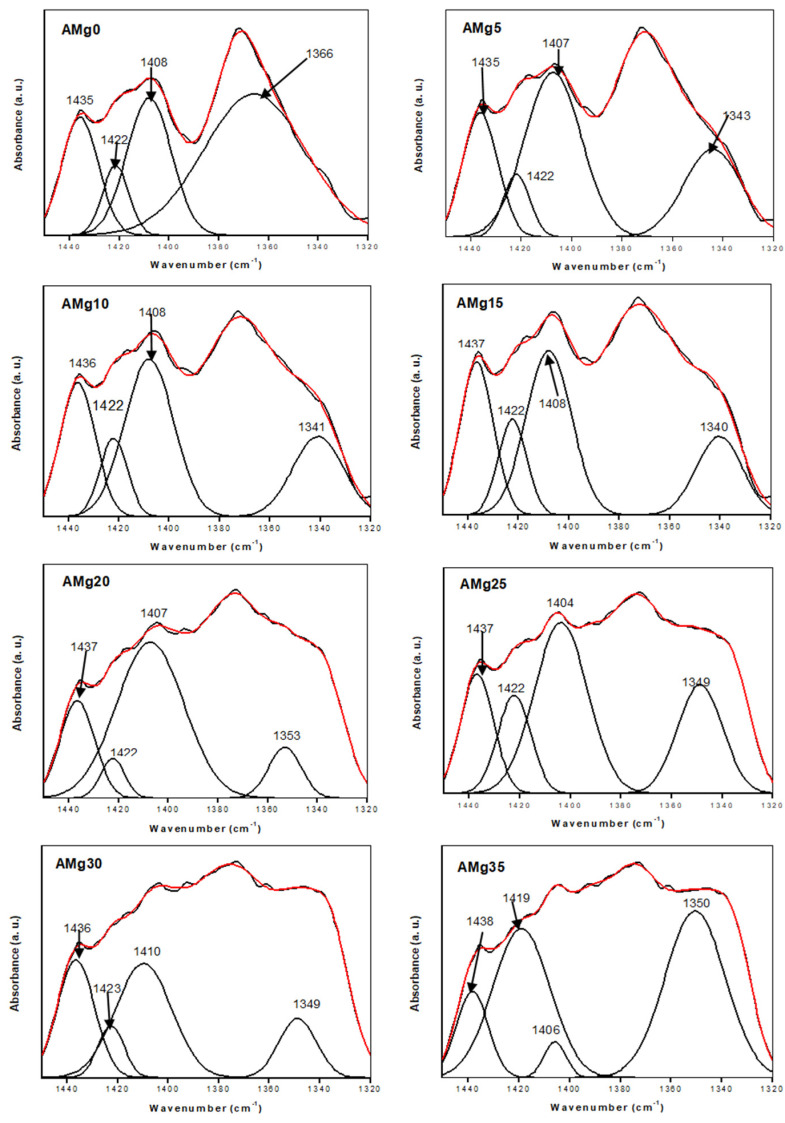
The deconvolution peak of FTIR spectra at a various weight percentage of Mg(NO_3_)_2_ doped in agarose–based gel polymer electrolyte at wavenumber range 1320 cm^−1^ to 1450 cm^−1^.

**Figure 6 polymers-13-03357-f006:**
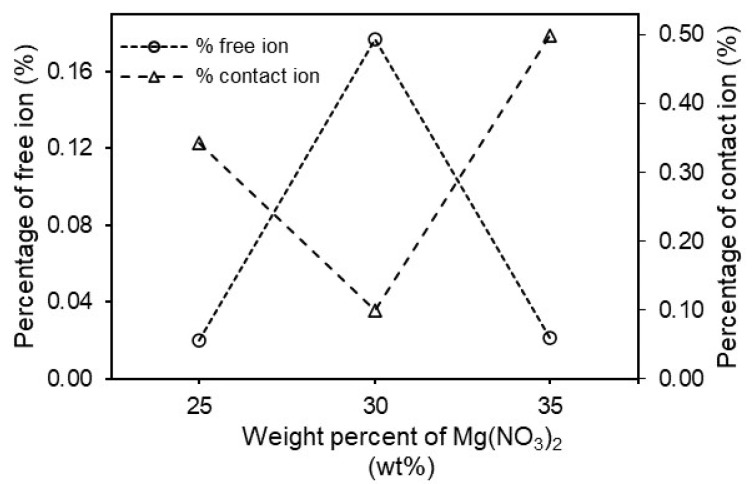
Graph of the percentage of free ions and contact ions for AMg25, AMg30 and AMg35 polymer electrolytes.

**Figure 7 polymers-13-03357-f007:**
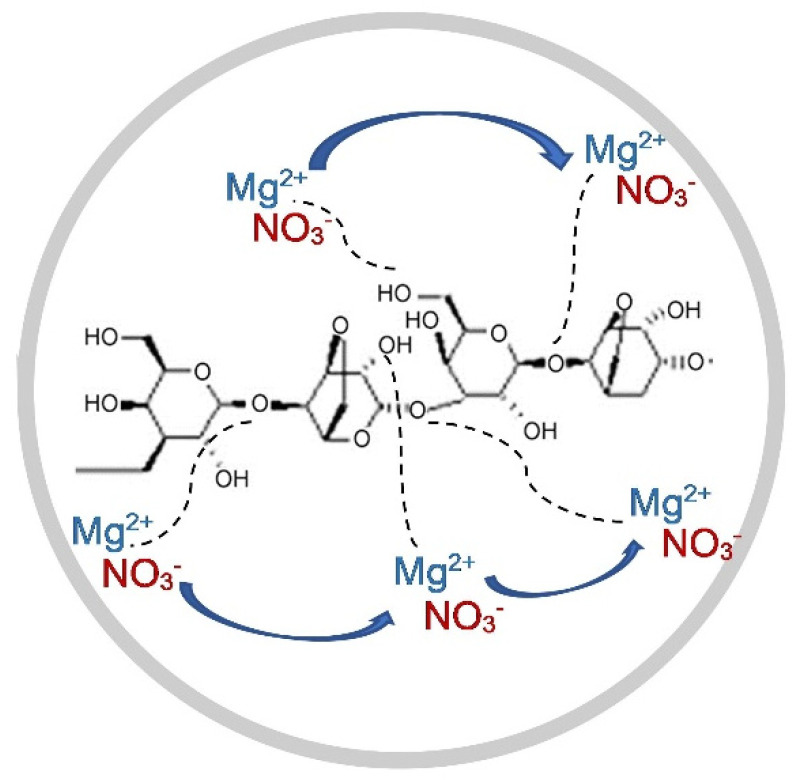
Mechanism of complexation between agarose molecules and Mg(NO_3_)_2_.

**Figure 8 polymers-13-03357-f008:**
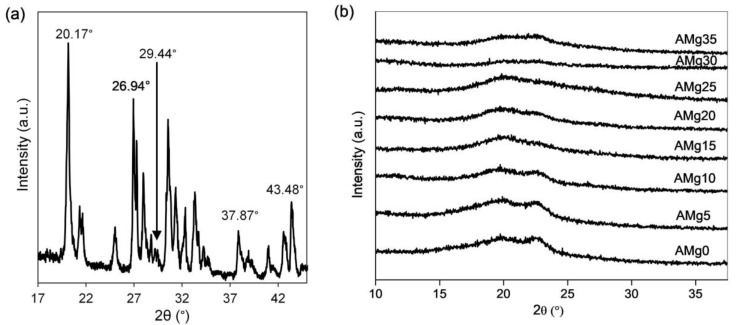
XRD patterns of (**a**) Mg(NO_3_)_2_ in range of 2θ = 17°−45° and (**b**) various weight percentage of Mg(NO_3_)_2_ incorporates into agarose–based polymer electrolyte between 2θ = 10°−37°.

**Figure 9 polymers-13-03357-f009:**
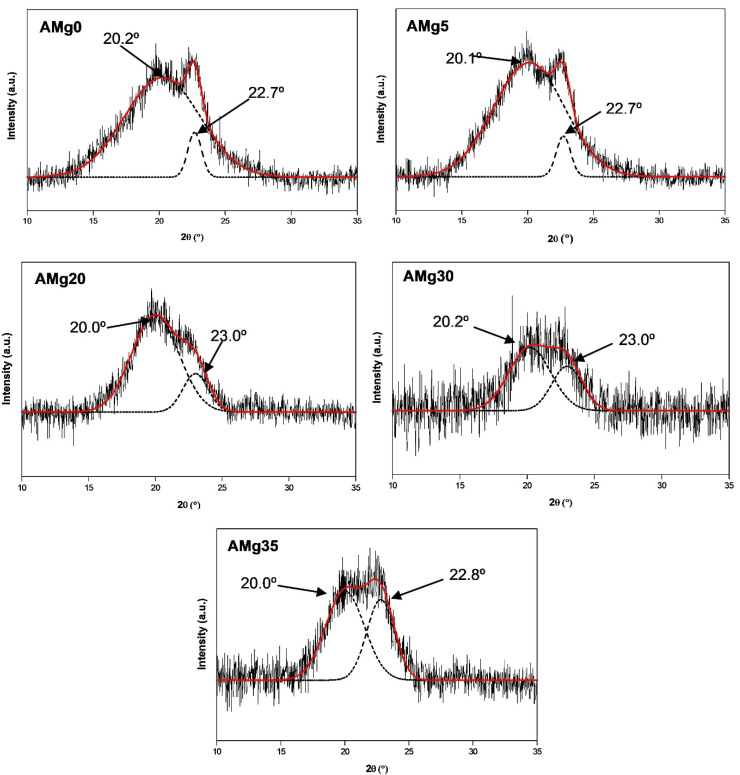
XRD deconvolution for different concentrations of agarose–Mg(NO_3_)_2_ polymer electrolytes.

**Figure 10 polymers-13-03357-f010:**
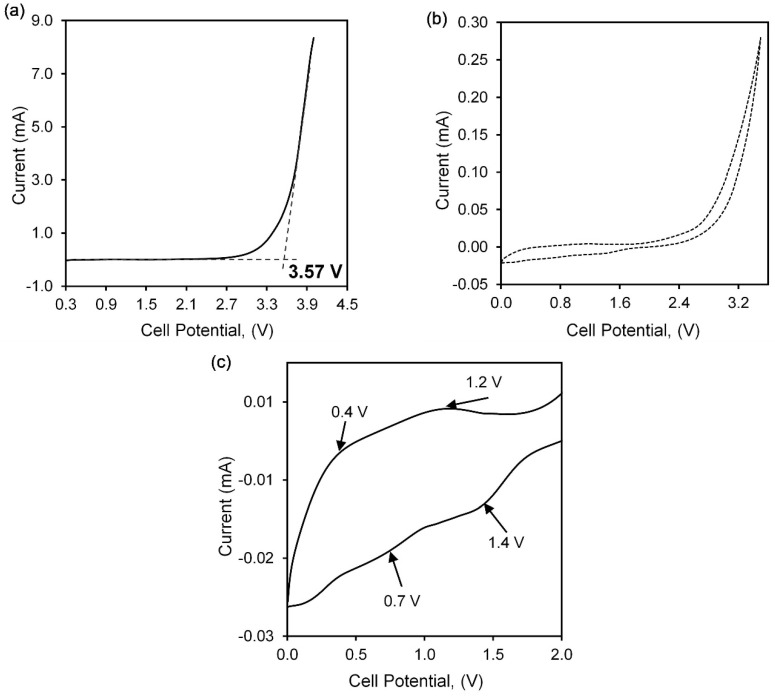
Graph of (**a**) linear sweep voltammetry, (**b**) cyclic voltammetry in range 0 to 3.5 V and (**c**) peaks observed in CV pattern of sample AMg30 polymer electrolyte.

**Table 1 polymers-13-03357-t001:** The degree of crystallinity, $X_{c}$ and size of crystallite, *L* of selected agarose–Mg(NO_3_)_2_ polymer electrolyte samples.

Mg(NO_2_)_3_ Content (wt%)	Degree of Crytallinity, *X_c_* (%)	*FWHM* (rad)	Crystalline Size, *L* (nm)
**0**	38.58	0.0214	**6.62**
**5**	36.69	0.0225	**6.28**
**20**	29.72	0.0392	**3.61**
**30**	11.29	0.0468	**3.03**
**35**	37.94	0.0446	**3.17**

## Data Availability

Not applicable.

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
