# Peer review of "Role of Mg(NO3)2 as Defective Agent in Ameliorating the Electrical Conductivity, Structural and Electrochemical Properties of Agarose–Based Polymer Electrolytes"

_polymers, 2021, doi:10.3390/polym13193357_

Round 1

Reviewer 1 Report

In this work, the agarose/Mg(NO3)2 polymer electrolyte has been fabricated by authors via solution mixing and subsequent the solution casting methods. The EIS, ionic conductivity, intermolecular interactions and electrochemical performances were characterized in detail. However, the following points should be clarified before publication and my opinion could be considered for publication after a major revision or rejection in this version: 1. There is a bunch of similar works for the investigation of polymer electrolytes. So what is new of this work? 2. Some grammar mistakes still exist in the main text, please revise is carefully. 3. As author mentioned, the conductivity of i-carrageenan-Mg(NO3)2 and polyethylene oxide (PEO)-Mg(NO3)2 are 6.10 × 10-4 S and 1.34 × 10-5 S.cm-1, respectively, which is much higher than this wok (1.48 × 10-5 S.cm-1). What is the advantage of this paper? The different or new point should provide in the introduction or abstract comparing with previous similar work. 4. For the Figure 1a to 1g, please do not separate it. Put them together, than show what is it in the caption. 5. All equations for characterizing the performances of electrolytes is from the previous work, so don’t need to show it all of them. If something new, you can show it. 6. Some new insights or mechanism should provide, not just show what we can see from the figures.

Author Response

Response to the reviewer’s comments as attached (uploaded document).

Reviewer 2 Report

The experimental work concerns the physical chemistry properties of the Agarose-Mg(NO3)2 polymer material. However, the discussion of the XRD data is insufficient.

1) The Full Width at Half Maximum (FWHM) must be quantified and the Scherrer equation applied.

2) In the pure-agarose XRD spectrum a single amorphous halo is observed due to pseudo-exagonal chain correlation. The double halos observed in the other cases indicate a more complex correlation in the amorphous phase, rather than crystalline reflections. Likely, these are the amorphous zones with different maximum of the inter-chain radial function due to the different concentration of the adsorbed ions.

In light of these considerations, the XRD analysis must be rewritten.

I point out two minor typos: ‘petri dish’ instead of ‘Petri dish’; FHWM instead of FWHM.

Author Response

(The authors gave the same response as above.)

Round 2

Reviewer 1 Report

Authors have revised all points as comments mentioned. This work should have some benefits for this research field. So, it should be accepted.

Reviewer 2 Report

The authors have accepted the suggestions of the referees: the work is now suitable for publication. I would like to point out some minor typos: Mg(NO3)2 (in Abstract) instead of Mg(NO3)2; S.cm-1 instead S ⋅ cm-1; wt.% instead of wt%.